# Antibacterial Activity and Protection Efficiency of Polyvinyl Butyral Nanofibrous Membrane Containing Thymol Prepared through Vertical Electrospinning

**DOI:** 10.3390/polym13071122

**Published:** 2021-04-01

**Authors:** Wen-Chi Lu, Ching-Yi Chen, Chia-Jung Cho, Manikandan Venkatesan, Wei-Hung Chiang, Yang-Yen Yu, Chen-Hung Lee, Rong-Ho Lee, Syang-Peng Rwei, Chi-Ching Kuo

**Affiliations:** 1Research and Development Center of Smart Textile Technology, Institute of Organic and Polymeric Materials, National Taipei University of Technology, Taipei 10608, Taiwan; wcl2320@mail.lit.edu.tw (W.-C.L.); duckfat8@gmail.com (C.-Y.C.); manikandanchemist1093@gmail.com (M.V.); f10714@ntut.edu.tw (S.-P.R.); 2Department of Applied Cosmetology, Lee-Ming Institute of Technology, New Taipei City 243083, Taiwan; 3Department of Chemical Engineering, National Taiwan University of Science and Technology, Taipei 10607, Taiwan; whchiang0102@gmail.com; 4Department of Materials Engineering, Ming Chi University of Technology, New Taipei City 24301, Taiwan; yyyu@mail.mcut.edu.tw; 5Division of Cardiology, Department of Internal Medicine, Chang Gung Memorial Hospital-Linkou, Chang Gung University College of Medicine, Tao-Yuan 333, Taiwan; 6Department of Chemical Engineering, National Chung Hsing University, Taichung 402, Taiwan; rhl@dragon.nchu.edu.tw

**Keywords:** electrospinning, nanofibrous membrane, antibacterial activity, protection efficiency, mask

## Abstract

Human safety, health management, and disease transmission prevention have become crucial tasks in the present COVID-19 pandemic situation. Masks are widely available and create a safer and disease transmission–free environment. This study presents a facile method of fabricating masks through electrospinning nontoxic polyvinyl butyral (PVB) polymeric matrix with the antibacterial component Thymol, a natural phenol monoterpene. Based on the results of Japanese Industrial Standards and American Association of Textile Chemists and Colorists methods, the maximum antibacterial value of the mask against Gram-positive and Gram-negative bacteria was 5.6 and 6.4, respectively. Moreover, vertical electrospinning was performed to prepare Thymol/PVB nanofiber masks, and the effects of parameters on the submicron particulate filtration efficiency (PFE), differential pressure, and bacterial filtration efficiency (BFE) were determined. Thorough optimization of the small-diameter nanofiber–based antibacterial mask led to denser accumulation and improved PFE and pressure difference; the mask was thus noted to meet the present pandemic requirements. The as-developed nanofibrous masks have the antibacterial activity suggested by the National Standard of the Republic of China (CNS 14774) for general medical masks. Their BFE reaches 99.4%, with a pressure difference of <5 mmH_2_O/cm^2^. The mask can safeguard human health and promote a healthy environment.

## 1. Introduction

Particulate matter (PM) pollution, due to industrial development and living environment change, worsens air quality and causes severe health problems, such as respiratory diseases, cardiovascular diseases, and allergies [1]. In the recent years, various influenza and coronavirus diseases have become prevalent, causing harm to human health and large economic losses [2,3,4]. Masks can prevent droplets and PM from invading the human body and prevent respiratory infections. The types of masks include cotton masks, general medical masks, surgical masks, activated carbon masks, and N95 masks [5,6]. N95 masks have the highest ability to filter out PM with a ≤2.5-µm diameter (PM2.5), but because of their favorable adhesion, they can cause breathing problems and are thus unsuitable for long-term wear and general protection. To keep out epidemic virus droplets, PM with a ≤10-µm diameter (PM10), and some PM2.5, and thus protect the human body, surgical and medical masks must be worn correctly and be suitable for long-term wear.

General medical masks are made of polypropylene (PP), and their manufacturing method mainly involves spun bonding or melt blowing. The outer and inner layers of the mask are composed of PP spun-bond nonwoven fabric. The outer layer must repel water to prevent the penetration of blood, body fluids, and other potentially infectious substances. The inner layer is a face affinity layer, which can absorb moisture generated during breathing and thus keep the face dry and comfortable. The middle layer of the mask is a melt-blown nonwoven fabric layer, which is used to filter PM and bacteria. However, when the fibers of the middle layer are thick, the material’s efficiency in filtering very small particles is low and thus high efficiency and low impedance cannot be achieved. Therefore, polytetrafluoroethylene membranes [7] or polyvinyl butyral (PVB) can be used as the matrix for preparing nanofiber membranes through electrospinning. Nanofibers have small diameter, high porosity, and internal pores [8,9,10,11,12,13], so they have favorable connectivity and high air permeability [14,15,16,17,18], which are conducive to capturing ultrafine particles. Nanofibers thus have high efficiency in filtering PM [19,20]. Many electrospun fiber membrane types have been manufactured as media for air filtration; the most common are those made from uniform and monostructured nanofibers, including ultrafine nylon 6 fibers [21,22,23], polyethylene oxide nanofibers [24], alumina nanofibers [25], and polyester nanofibers [26,27,28,29]; however, these materials cannot filter bacteria or viruses in the air.

Thymol (2-isopropyl-5-methyl phenol) is a monoterpene phenol found in the essential oils of herbal plants [30]. Many studies have shown that Thymol has antibacterial and antifungal properties [31]. Marino et al. [32] reported that *Thymus vulgaris* L. essential oil has antibacterial activity against both Gram-negative bacteria (e.g., *Escherichia coli*) and Gram-positive bacteria (e.g., *Staphylococcus aureus*). By employing the agar dilution method, Nostro et al. [33] found that Thymol has antibacterial effects and sensitivity for various types of *S. aureus*. Kavoosi et al. [34] mixed gelatin films with different concentrations of Thymol to test its antibacterial activity against *E. coli*, *Pseudomonas aeruginosa*, *Bacillus subtilis*, and *S. aureus* for potential use as an antibacterial nano wound dressing.

Some scholars have blended Thymol with polymer materials to make nanofiber membranes and tried to develop dressings for wound healing, such as by adding Thymol to poly(ε-caprolactone) and polylactic acid through electrospinning technology to prepare nanofibrous mats [11,35]. Thymol has also been added in situ to chitosan/polyethylene glycol fumarate to prepare hydrogel wound dressings. A study demonstrated that hydrogels containing Thymol have favorable mechanical properties and excellent antibacterial activity against both Gram-negative and -positive bacteria, and they can be applied in a dressing for infected wounds with a moderate amount of exudate [36].

The main mechanisms through which fibrous materials filter dust and particles are the interception effect, inertial deposition, Brownian diffusion, the electrostatic effect, and the gravity effect. Effective filtration of dust and particles by the filter materials in a mask is achieved through a combination of these five effects [37]. In the present study, Thymol was mixed with PVB through electrospinning to prepare antibacterial nanofiber membranes. Antibacterial property and protection efficiency tests were conducted to ensure that the developed medical masks meet performance specifications and have antibacterial effects.

## 2. Experimental

### 2.1. Materials and Microorganisms

Polyvinyl butyral (PVB, B18-HX, *M*_W_∼100,000, Chang Chun Co., Taipei, Taiwan) was used as the matrix of the nanofibrous membranes. Ethanol (95%) was used as a solvent purchased from I-Chang Chemical Co. ROC. Thymol (5-Methyl-2-(propan-2-yl)phenol) was supplied by Scharlab TI0080 (99% purity). The antibacterial activities of Thymol-containing PVB nanofibrous mats were evaluated using *Staphylococcus aureus* (*S. aureus*) (ATCC 6538), *Klebsiella pneumoniae* (*K. pneumoniae*) (ATCC 4352), *Escherichia coli* (*E. coli*) (ATCC 8739), which were obtained from the Bioresource Collection and Research Center (BCRC), Taipei, Taiwan.

### 2.2. Preparation of PVB Antibacterial Nanofibrous Membrane Containing Thymol

Thymol/PVB nanofibrous membranes were fabricated by vertical electrospinning technique. Primarily, 5 wt% of PVB polymer solution was prepared by dissolving 25 g of PVB powder in 475 g of ethanol solvent with magnetic stirring for 2 h at room temperature and obtained transparent solution. Subsequently, different amounts of Thymol powder were added to the above mixture with vigorous stirring to obtain the homogenous solution, various compositions of Thymol/PVB electrospinning solutions as shown in Table 1. The prepared Thymol/PVB electrospun solution was loaded into the plastic syringe containing a stainless needle (22-gauge stainless). A syringe pump (KDS-200, KD Scientific., Holliston, MA, USA) was used to feed the electrospinning solution with a fixed feeding rate of 0.72 mL/h. A high-voltage power supply (SM3030-24P1R, YOU-SHANG TECHNICAL CORP) was employed, and the applied voltage was 27 kV. The distance from tip to the collector was fixed at 10 cm, and the collecting lasted for 7 h.

### 2.3. Preparation of Thymol/PVB Antibacterial Nanofibrous Masks

A result from the antibacterial preliminary test 0.6:1 ratio of Thymol/PVB nanofibrous membrane shows strong antibacterial effect against *S. aureus*, *K. pneumoniae*, and *E. coli*. Therefore, this optimized ratio was used in a vertical electrospinning device (SC-PME50, COSMI) to prepare the Thymol/PVB composite antibacterial nanofibrous mask. Since the vertical electrospinning device has many advantages, such as the needle can be moved left and right, collection area can be rotated, speed can be controlled, and a large-area nanofibrous membrane with better uniformity can be obtained. To this interest, the composite electrospun solution was taken into a 15 mL syringe, a stainless 22-gauge needle installed on the needle auxiliary moving mechanism. Relevant parameters were adjusted according to the experimental design, as shown in Appendix A. Figure 1 represents the basic work function of three-layer facemask.

### 2.4. Surface Observation of Nanofiber Membranes

A scanning electron microscope (SEM) (JSM-6510, JEOL, Tokyo, Japan) was used to analyze the surface morphology of inner, outer, and middle layers of commercial mask fabrics and different Thymol/PVB electrospun nanofibrous membranes with varying electrospinning parameters. Diameters of uniformly distributed nanofiber were calculated from the SEM image.

### 2.5. Antibacterial Activity

#### 2.5.1. Antibacterial Qualitative Tests

Two qualitative methods were used to evaluate the antibacterial activities of PVB nanofibrous membranes containing Thymol, described below:

Japanese Industrial Standards (JIS) L1902 (Halo method): antibacterial qualitative tests were determined with adherence to the procedure of JIS L1902 method [38,39] (Halo method), performed by observing the zone of inhibition to evaluate the antibacterial properties of textiles. The method is suitable for textiles process by leaching antibacterial agents. The test methods were explained as follows:

A known weight of test samples was added to observe whether the Petri dish had a zone of inhibition; we measured the width of the sample (D) and the sum of the width of the sample and the zone of inhibition (T); then calculated the width of the zone of inhibition (W) with the following formula (1), and the result is expressed as an average 0.1 mm.
W = (T − D)/2(1)

American Association of Textile Chemists and Colorists (AATCC) 147 method (Parallel Streak method): the antibacterial qualitative tests were carried out in accordance with the procedures of AATCC147 method (Parallel Streak method). This method also helps to evaluate the antibacterial properties of textiles and it is suitable for textiles process by leaching antibacterial agents. The testing process was explained as follows:

(1) CZ: Clear zone of inhibition

(2) I: Inhibition of growth under the sample only

(3) NI: No inhibition of growth

#### 2.5.2. Antibacterial Quantitative Test

JIS L1902 (Absorption method) testing method serves the quantitative information of antibacterial properties for the antibacterial textile products. The testing procedure was expressed as follows:

(1) The growth value on the control (F) should be ≥1.0.

F = log C_t_ − logC_o_(2)

log C_t_: The average logarithm of the number of bacteria after 18–24 h inoculation of the control sample

log C_o_: The average logarithm of the number of bacteria recovered from the control at the beginning of contact time

(2) Antibacterial activity value (A) should be ≥2.0, which means the sample has antibacterial effect, according to the descriptions of JIS L1902.
A = (log C_t_ − log C_o_) − (log T_t_ − log T_o_) = F − G(3)

G: The growth value on the sample

log T_t_: The average logarithm of the number of bacteria after 18–24 h inoculation of the sample

log T_o_: The average logarithm of the number of bacteria recovered from the sample at the beginning of contact time

#### 2.5.3. Particulate Filtration Efficiency (PFE)

Following the procedure of National Standards of the Republic of China (CNS) 14,755 [40], the Automated Filter Tester (Model 8130, TSI, Shoreview, Minnesota, MN, USA) was used to measure the submicron particulate filtration efficiency or penetration efficiency of the masks to understand the protection performance of the masks. The definition of submicron particulate filtration efficiency as follows:

PFE: The ratio of the aerosol concentration captured by the mask to the original upstream aerogel concentration.
Penetration efficiency (%) = (aerosol concentration through the mask)/(upstream air aerosol concentration) × 100%Protection efficiency (%) = 100 − penetration efficiency (%)(4)

Inhalation resistance: the ventilation resistance generated by a certain flow of air in the inhalation direction of the mask.

#### 2.5.4. Differential Pressure of Air Exchange

According to CNS 14,777 (Test Method for Air Exchange Pressure of Medical Masks) Section 3, the tester of air exchange pressure (TTRI, TW) was used to measure the differential pressure of air exchange of the mask, to understand the breathability of the mask, and to evaluate whether it will cause breathing difficulties when wearing.

#### 2.5.5. Bacterial Filtration Efficiency (BFE)

Following the procedure of CNS 14775, the Bacterial Filtration Efficiency (TTRI, TW) of the synthesized mask was calculated. A formed bacteria (*S. aureus* biological) aerogel size is approximately (3.0 ± 0.3) μm, which is controlled by the nebulizer system. The percentage of bacteria absorption before and after filtering through the mask and rate of applied pressure was estimated. The results were subjected to the following equation to calculate the percentage of BFE. The same procedure was followed for the 30 commercial masks to compare the obtained results.
BFE (%) = 100 × (C − T)/C(5)

C: Average number of total plate count on the control.

T: Total plate count on the sample.

Calculate the average particle size of the aerosol by formula (6), which should be (3.0 ± 0.3) μm.
Average particle size (MPS) = Σ(An×Sn)/ΣAn = (A1 × 7.0 + A2 × 4.7 + A3 × 3.3 + A4 × 2.1 + A5 × 1.1+ A6 × 0.65)/(A1 + A2 + A3 + A4 + A5 + A6)(6)

An: The number of bacteria in the petri dish of this stage

Sn: The particle size of the aerosol collected by the sampler at this stage

## 3. Results and Discussion

### 3.1. Antibacterial Qualitative Analysis for Nanofibrous Membrane of Thymol/PVB Blenders

Antibacterial qualitative analysis of the nanofibrous membranes was performed using the JIS L1902 and AATCC 147 standardized methods. Although no inhibition zones were found in samples of different compositions, to improve the inhibition zones, various methods were employed, such as adding a surfactant [41] and increasing the contact opportunities for bacterial liquid with nanofiber membrane (by using nets, pressurized glass sheets, and alumina foil). The addition of a surfactant can affect nanofiber surface morphology [42]. *S. aureus* were used as a testing agent in the JIS L1902 and AATCC 147 qualitative methods to evaluate the antibacterial effects of Thymol/PVB nanofibrous membranes. Although, as shown in Table 2 and Appendix A, spraying the tested bacteria liquid on the nanofibrous membrane or net-like architecture did not form any contact with the substrate. Even applying gentle pressure using a glass plate, this lack of interface formation results to fail the formation of the inhibition zone. In contrast, Group A and Group C showed the formation of zone inhibition on alumina foil with a high concentration ratio of Thymol in PVB (1:1). However, even though the width of the inhibition zone increased slightly, the difference was not large.

The inhibition zone may not have appeared because Thymol and PVB are insoluble in water, meaning that the probability of Thymol coming into contact with the inoculum was low. The weight of the sample in the antibacterial film was 4.76 times that in the nanofibrous membrane; that is, the content of the nanofiber membrane per unit area was low (Appendix A). The nanofibrous membrane had a network structure, and actual contact with the bacterial liquid only occurred on the fiber mesh surface; however, the inhibition zone could only be produced when the antibacterial substance was in uniform contact with the bacterial liquid. For these reasons, the zone of inhibition was difficult to observe when the nanofibrous membranes were tested using the qualitative method.

### 3.2. Antibacterial Quantitative Analysis for Thymol/PVB Nanofibrous Membranes

The results in 3.1 showed that the qualitative method was not suitable for evaluating the antibacterial activity of Thymol/PVB nanofibrous membranes. Therefore, a quantitative determination was performed based on Absorption method from JIS L1902 to analyze the antibacterial activity of Thymol/PVB nanofibrous membranes on Gram-positive (*S. aureus*)*,* and Gram-negative (*K. pneumoniae* and *E. coli*) bacteria.

#### 3.2.1. Staphylococcus aureus

The antibacterial activity of the Thymol/PVB nanofibrous membranes on *S. aureus* is displayed in Appendix A and Figure 2a. According to the description of JIS L1902, when the antibacterial activity value exceeds 2.0, the samples are considered to have antibacterial properties [43,44,45,46]. With the addition of Thymol to PVB, the bacterial growth was considerably reduced. For example, considering Thymol/PVB = 0.2:1 nanofiber with an antibacterial value of 2.8, this increment is attributed to the phenolic hydroxy functional group of Thymol. Consequently, a maximum amount of antibacterial value 6.4 was obtained from the optimized ratio of 0.6:1. However, further addition of Thymol does not cause any changes in it and clearly shows the saturated ratio to PVB. Therefore, enhancing the proportion of Thymol in PVB leads to increasing antibacterial activity toward *S. aureus* bacteria.

#### 3.2.2. Klebsiella pneumoniae

The study of antibacterial effects of Thymol/PVB nanofibrous membranes against *K. pneumoniae* were monitored for 18–24 h. The control group growth value (F) should have been >1.0 and the test conditions were established. Appendix A and Figure 2b display that bacteria growth was controlled as the proportion of Thymol increased. Antibacterial activity value also reached a maximum of 6.4 when the ratio of Thymol: PVB was 0.4:1. Other proportions also provide the same result.

#### 3.2.3. Escherichia coli

As shown in Appendix A and Figure 2c, the quantitative antibacterial effect of the various nanofibrous membranes against *E. coli* was in favorable agreement with the testing results for other bacteria. However, the Thymol/PVB (0.6:1) nanofibrous membrane exhibited considerable enhancement, with a value of 6.4. Therefore, enhancing the proportion of Thymol in PVB resulted in higher antibacterial activity.

A comparison of the antibacterial activity values of PVB and various-ratio Thymol/PVB nanofibrous membranes for three bacterial types is presented in Appendix A. The PVB nanofiber membrane without Thymol exhibited no response to the bacterial colonization. A significant reduction in bacterial growth was discovered for the Thymol/PVB nanofiber membranes, and the reduction depended on the concentration of Thymol. Even at a low Thymol:PVB ratio of 0.2:1, the obtained antibacterial activity value against the Gram-positive *S. aureus* bacteria was 2.8, which indicated that this membrane had relevant antibacterial activity. By contrast, the antibacterial activity values of this membrane against the Gram-negative *K. pneumoniae* and *E. coli* bacteria were only 1.6 and 0.7, respectively. According to the JIS L1902 description, the membrane thus had no relevant antibacterial activity. Therefore, the growth of inhibition was observed for Gram-positive bacteria even at a low concentration of Thymol as compared with the Gram-negative bacteria. This variation was also reported in previous research [34]. The Gram-negative bacteria were protected by cell-wall lipopolysaccharides and outer membrane proteins, which restrict the diffusion of hydrophobic compounds through the lipopolysaccharide layers. However, at a higher concentration of antimicrobial agents, the polysaccharide layer was destroyed by essential oils.

As illustrated in Figure 2d, the sample with ratio 0.6:1 had the highest activity, indicating that the bacterial counts of the three test bacteria after culture were all <20 colony-forming units (CFUs), meaning that all the bacteria died. Furthermore, when the Thymol:PVB ratio was increased to 0.8 and 1, the antibacterial activity value did not change. Therefore, the Thymol:PVB ratio 0.6:1 (*w*:*w*) was the most suitable ratio for preparing the antibacterial nanofiber mask.

#### 3.2.4. Diameter of Nanofibrous Membrane for Thymol/PVB Blenders

As shown in Figure 3 and Figure 4, Thymol/PVB nanofiber membranes prepared with different proportions had an average fiber diameter between 500 and 700 nm, which belonged to the broad range of nanofibers.

### 3.3. Analysis of the Protection Efficiency of Antibacterial Nanofiber Masks

#### 3.3.1. Submicron Particulate Filtration Efficiency (PFE) and Pressure Difference of Air Exchange

The optimized ratio (0.6:1) Thymol/PVB antibacterial nanofiber masks were prepared using a vertical electrospinning device, and protection efficiency and pressure difference tests were conducted in accordance with the CNS 14,755 and CNS 14,777 standards. The electrospinning parameters employed are presented in Appendix A, and the results are shown in Table 3. Groups C and D collected nanofiber webs at 25 kV and 5 mL/h for 45 min, and although the collection distance was different for these groups, the PFE was >80% for both. However, the pressure difference exceeded the 5 mmH_2_O/cm^2^ threshold specified by CNS 14,774 for surgical masks, indicating that the mask would cause breathing problems during wear. According to the parameter settings in Appendix A, the influences of voltage and flow rate on the PFE, impedance, and pressure difference were explored. In Group E, electrospinning was performed at the voltage of 15 kV and flow rate of 2 mL/h. The test results revealed that the PFE was 82.7% and pressure difference was 4.5 mmH_2_O/cm^2^, which meets CNS 14,774 standard requirements for surgical masks.

#### 3.3.2. The Fiber Diameter and PFE

The fiber diameters were measured by SEM. Appendix A and Table 4 exhibit that when the flow rate was reduced from 5 mL/h to 2 mL/h, the average fiber diameter was reduced from 836 ± 329 nm to 375 ± 69 nm. This result showed that as the flow rate decreased, the fiber diameter greatly decreased and became more uniform, and could effectively improve the impedance value and the high-pressure difference. For the nanofiber mask, the thinner and denser the fiber diameters are, the more PFE and pressure difference can meet the requirements of CNS 14,774 specifications for general medical masks [47].

In the construction of general commercial masks, PP spun-bond is used as the outer and inner layers with a diameter of 20~30 μm. The middle layer is a melt-blown nonwoven fabric layer with 2–10 μm diameters. The fiber diameter of each nonwoven fabric was observed by SEM (Appendix A). However, most of the airborne viruses (Corona) spread through the aerosol are in nanometers [48,49,50,51]. The fiber diameter of the nanofiber prepared in this study was 375 + 69 nm, and the filter efficiency was improved by reducing the fiber diameter. Hence, the antibacterial nanofibers are well replaceable for the nonwoven fabric layers.

#### 3.3.3. Spinning Parameters and PFE

Although the electrospinning parameters of Group E can meet the specifications of CNS 14,774 mask, at 15 kV, the spinning solution was not able to form a stable tailor cone due to the droplets block in the needle nozzle. To overcome this, the applied voltage was increased to 18 kV (Group F) and we found that the formation of beadles continuous nanofiber. The Group F parameters are given in Appendix A (18 kV, distance 160 mm, 2 mL/h), under different spinning times. Correspondingly, the influence of PFE was further explained in Appendix A, Figure 5, that as the spinning time increased, the accumulated nanofiber membranes increased. When the collection time reached 6 h, the PFE increased from 38.6% to 83.2%, and the pressure difference was fixed at 4.7 mmH_2_O/cm^2^, which meets the requirements of CNS 14,774 [47].

#### 3.3.4. Bacteria Filtration Efficiency (BFE)

In Taiwan, general medical masks must meet the specifications of CNS 14774, which state that the BFE must be >95%. From the viewpoint of air permeation, the BFE and PFE of a filter material should be correlated. Therefore, in this study, 30 commercial masks were tested and analyzed to determine the correlation between their BFE and PFE. In addition, a Thymol/PVB nanofibrous membrane was prepared through electrospinning with the Group F parameters, with fiber collection lasting 1–6 h, and it was observed that increasing the nanofiber collecting time leads to the higher areal density nanofiber membrane, which helps to trap the bacteria aerosols. Furthermore, these BFE results were compared with the PFE data. As shown in Appendix A, the correlation between BFE and PFE for the different masks were nonlinear, although when the PFE exceeded 73%, the BFE reached its maximum. Therefore, in this study, we prepared the mask with the aim of achieving the PFE of ≥80% and then conducted the BFE test. In theory, if the BFE value exceeded 95%, it could meet the CNS 14,774 specifications.

To demonstrate this, a BFE test was performed to the Group F nanofiber membrane by varying its electrospinning parameters. The results are shown in Figure 6 and Appendix A. The PFE of the antibacterial nanofiber mask was 83.2%, and its BFE was 99.4%, which was consistent with the correlation in the data analysis, as shown in Figure 6h. Moreover, to support this, many other researchers also have investigated the filtration efficiency using the analytical methods on the porous fiber membranes [52,53].

## 4. Conclusions

An extension of industries may address their countries’ growth, although it brings an inevitable side effect to their people. Especially, food and airborne diseases cause some serious issues. The face masks are an affordable and effective safety measure. Cost-effective bio-degradable eco-friendly masks are an inviting field of research due to increasing demands on healthcare and rising concerns of personal hygiene and safety. Through this research, we opened up a low-cost biodegradable face mask fabrication process, which can protect and sustain the human life.

In this research, a Thymol/PVB composite nanofibrous membrane was produced with various optimization conditions for the antibacterial mask. As prepared, the Thymol: PVB = 0.6:1 composite nanofibrous mask shows better performance, which meets the specifications of CNS 14774. Especially, bacterial filtration efficiency and pressure difference exhibit better performance than the commercial mask. Quantitative antibacterial activity test with JIS L1902 absorption method shows that as the content of Thymol increased, the antibacterial activity of Thymol/PVB nanofibrous membranes increased. The antibacterial activity values against Gram-positive bacteria of *S. aureus* and Gram-negative bacteria of *K. pneumoniae* and *E. coli* were 5.6 and 6.4, respectively. PFE values mainly depend on the electrospinning parameters. At 18 kV, 2 mL/h, the PFE met 83.2%, the inspiratory impedance was 10.8 mmH_2_O/cm^2^, and the pressure difference was 4.7 mmH_2_O/cm^2^, which fixes the respiration problem of the medical mask. BFE was 99.4%, which also verified the correlation between BFE and PFE data analysis inferred in this study. That is, when PFE was >73%, BFE reached ≥99% or more. Based on the results, Thymol/PVB nanofibers can be an alternate membrane layer for this pandemic environment, which is expected to be used as a multifunctional antibacterial mask against bacteria and viruses.

## Figures and Tables

**Figure 1 polymers-13-01122-f001:**
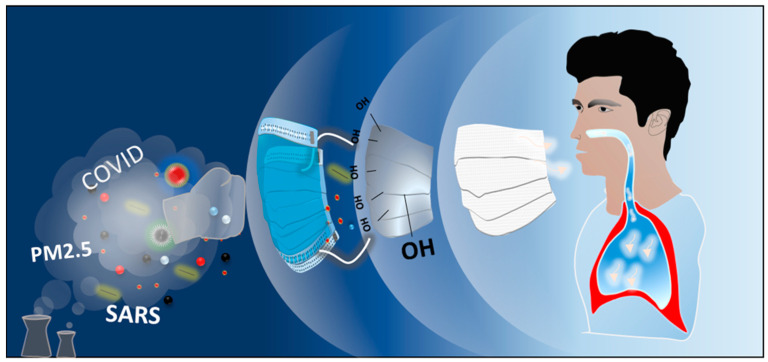
Schematic diagram of the electrospun Thymol/PVB nanofibrous facemask.

**Figure 2 polymers-13-01122-f002:**
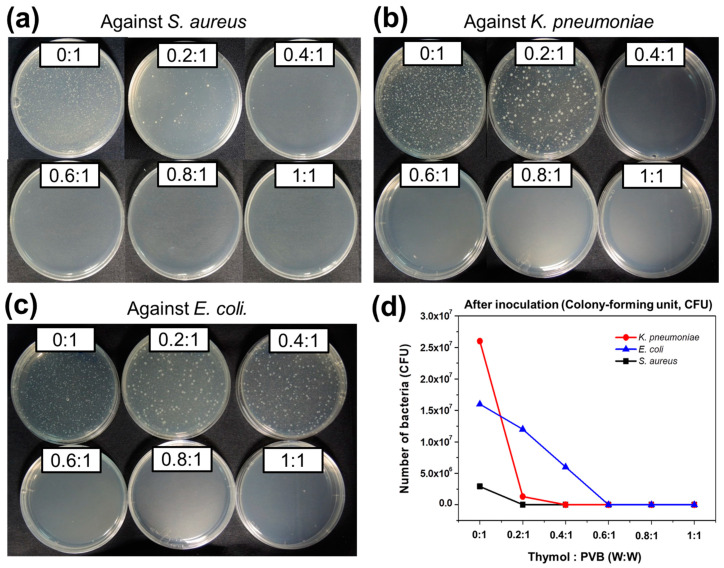
Quantitative results of antibacterial activity of Thymol/PVB nanofibrous membranes against (**a**) *S**taphylococcus*
*aureus*, (**b**) *K**lebsiella pneumonia*, (**c**) *E**scherichia coli.* (**d**) Comparison of the bacterial counts after culturing the three bacteria with Thymol/PVB fibrous membranes.

**Figure 3 polymers-13-01122-f003:**
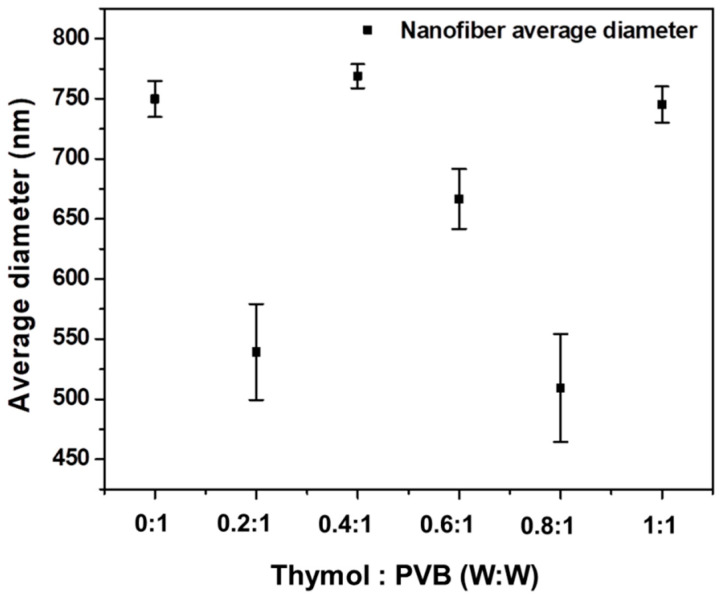
Diameter of nanofibrous membranes for Thymol/PVB blenders.

**Figure 4 polymers-13-01122-f004:**
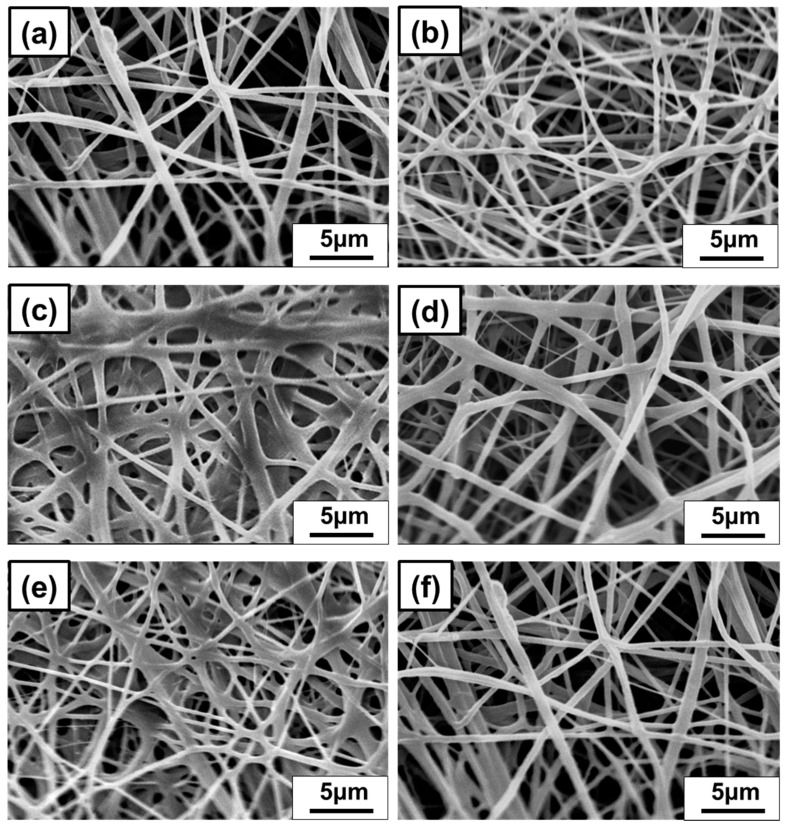
Nanofibrous membrane of Thymol/PVB blenders SEM pattern. (**a**) Thymol:PVB = 0:1, (**b**) Thymol:PVB = 0.2:1, (**c**) Thymol:PVB = 0.4:1, (**d**) Thymol:PVB = 0.6:1, (**e**) Thymol:PVB = 0.8:1, (**f**) Thymol:PVB = 1:1.

**Figure 5 polymers-13-01122-f005:**
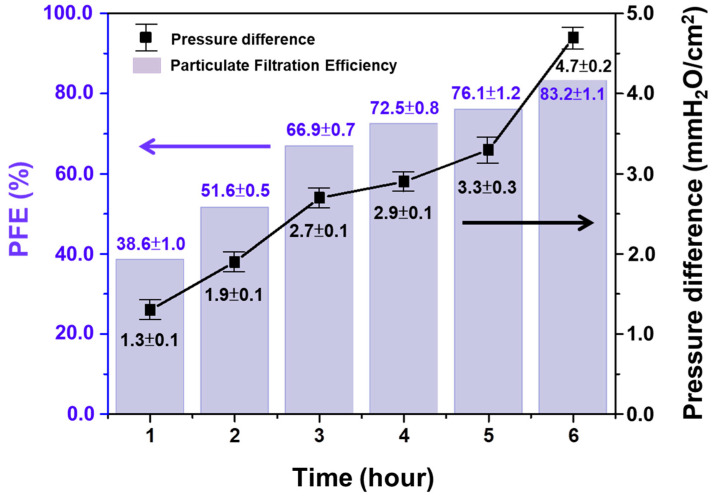
Testing results of particulate filtration efficiency (PFE) and pressure difference.

**Figure 6 polymers-13-01122-f006:**
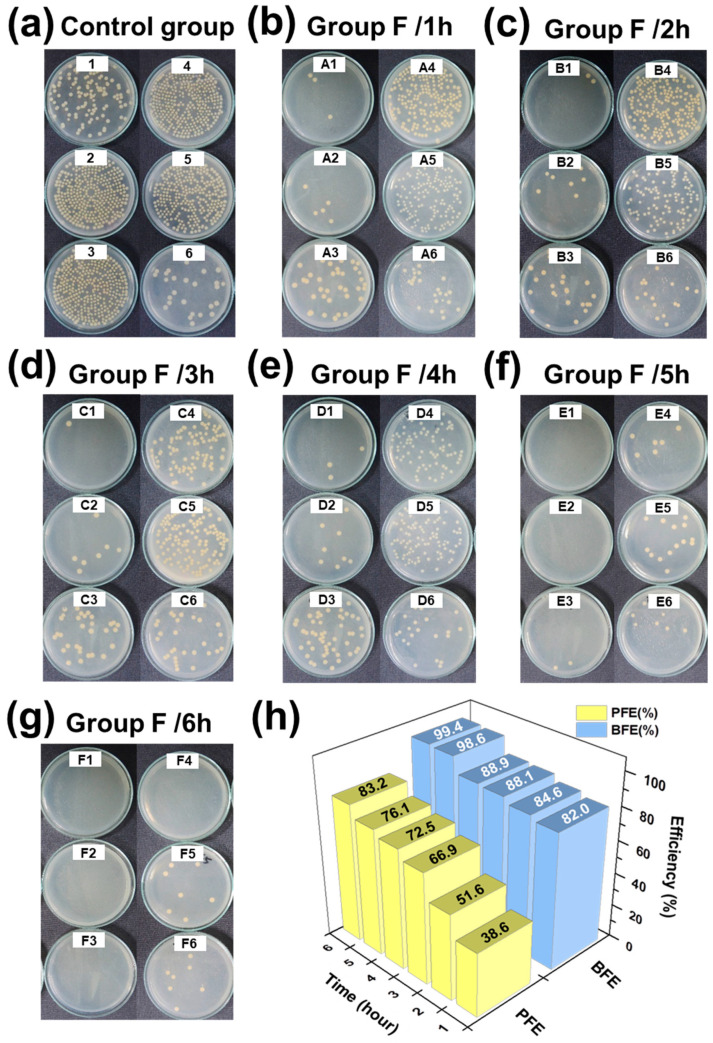
Bacterial filtration efficiency (BFE) test results of Group F spinning at different collection times. (**a**) Control group, (**b**) Group F/1h, (**c**) Group F/2h, (**d**) Group F/3h, (**e**) Group F/4h, (**f**) Group F/5h, (**g**) Group F/6h. (**h**) Correlation results of PFE and BFE tests in Group F.

**Table 1 polymers-13-01122-t001:** Various ratios of the Thymol/polyvinyl butyral (PVB) electrospinning solutions.

Thymol:PVB (*w*:*w*)	Thymol (g)	PVB (g)	5 wt% PVB (g)
0:1	0	2.5	50
0.2:1	0.5	2.5	50
0.4:1	1	2.5	50
0.6:1	1.5	2.5	50
0.8:1	2	2.5	50
1:1	2.5	2.5	50

**Table 2 polymers-13-01122-t002:** Qualitative analysis of antibacterial properties of Thymol/PVB nanofibrous membranes with different proportions.

Method		JIS L1902 (Halo Method)		AATCC147 (Parallel Streak Method)	
Carrier		Aluminum Foil	Net	Aluminum Foil	Net
Pressing with glass		none	yes	yes	yes
	group	A	B	C	D
Thymol:PVB (*w*:*w*)					
0:1		^c^ NI	^c^ NI	^c^ NI	^c^ NI
0.2:1		^c^ NI	^c^ NI	^c^ NI	^c^ NI
0.4:1		^c^ NI	^c^ NI	^c^ NI	^c^ NI
0.6:1		^c^ NI	^c^ NI	^c^ NI	^c^ NI
0.8:1		^a^ CZ 2.0 mm	^c^ NI	^a^ CZ 2.1 mm	^c^ NI
1:1		^a^ CZ 5.1 mm	^c^ NI	^a^ CZ 6.4 mm	^c^ NI

^a^ CZ: Clear zone of inhibition, ^c^ NI: No inhibition of growth.

**Table 3 polymers-13-01122-t003:** Results of protection efficiency in different electrospinning parameters.

Group	Spinning Time (min)	CNS 14755		CNS 14777
		PFE (%)	Inspiratory Impedance (mmH_2_O)	Pressure Difference (mmH_2_O/cm^2^)
C	0	32.8 ± 2.5	3.9 ± 0.3	1.4 ± 0.2
C	15	65.0 ± 2.5	9.1 ± 0.9	3.8 ± 0.1
C	30	67.1 ± 2.0	11.4 ± 0.6	4.4 ± 0.1
C	45	89.5 ± 1.0	38.7 ± 0.8	13.7 ± 1.8
C	60	94.2 ± 0.3	54.2 ± 8.6	14.1 ± 4.5
C	120	89.5 ± 1.6	42.7 ± 4.9	12.6 ± 2.0
D	0	7.5 ± 0.1	1.8 ± 0.1	0.8 ± 0.1
D	45	85.1 ± 0.7	29.9 ± 0.4	9.7 ± 0.1
E	60	62.9 ± 1.2	7.3 ± 0.2	2.6 ± 0.2
E	120	82.7 ± 1.0	12.3 ± 0.4	4.5 ± 0.3

**Table 4 polymers-13-01122-t004:** Comparison of fiber diameters in the middle layer of nanofiber masks.

Group/Spinning Time	D/45 min	E/2 h
Voltage	25 kV	15 kV
Flow rate	5 mL/h	2 mL/h
PFE(%)	85.1	82.7
Inspiratory impedance (mmH_2_O)	29.9	12.3
Pressure difference (mmH_2_O/cm^2^)	9.7	4.5
Average diameter (nm)	836	375
Standard deviation	329	69

## Data Availability

The data presented in this study are available on request from the corresponding author.

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
