# Peer review of "Antibacterial Activity and Protection Efficiency of Polyvinyl Butyral Nanofibrous Membrane Containing Thymol Prepared through Vertical Electrospinning"

_polymers, 2021, doi:10.3390/polym13071122_

Round 1
Reviewer 1 Report
The authors have fabricated thymol/PVB membrane and demonstrated its use for face masks. The novelty of this study is not clear. There are many articles that can be found on antibacterial membranes and fibers prepared using electrospinning. The authors must clearly specify how this study is different and how it is novel. Why PVB was chosen? Why was Thymol used and not silver particles?
- The last column in Table 1 is confusing. What does 5wt % PVB (g) mean? Do the authors mean to say that this is the overall weight of the solution?
- Figure 1 can be improved. I would recommend to only include the construct necessary for the mask and exclude the other things in the image. For example, the image implies that the mask can even stop covid virus. However, no evidence is provided. The size of the virus is much smaller than the bacteria.
- Section 2.3 can be revised. The advantages presented for vertical electrospinning is not unique to vertical setup. It can also be possible for horizontal setup.
- Where was the surfactant added? How was it added and how much of it was added?
Author Response
We appreciate the comments from the Academic editor and reviewers. The following is our point-to-point response to the comments.
Review-1
The authors have fabricated thymol/PVB membrane and demonstrated its use for face masks. The novelty of this study is not clear. There are many articles that can be found on antibacterial membranes and fibers prepared using electrospinning. The authors must clearly specify how this study is different and how it is novel. Why PVB was chosen? Why was Thymol used and not silver particles?
Answer: Thanks for the reviewer's deep consideration of our manuscript. The N95 commercial medical mask offers good bacteria filtration efficiency than the others although they cannot be wear for a long time because of their favourable adhesion property. Moreover, melt-blown fabrics diameter range is approximately <2 m (Table S8, Figure S2) which is not suitable for very tiny particles because, if fiber diameter is high the accumulation of fiber at a specific area would be less. Therefore, we employ facile electrospinning process to optimize the fiber diameter to meet CNS 14774 specification for filtration. We also estimated air penetration test in order to avoid breathing difficulties. The hydrophobic property of PVB not only protect us from water droplet environment it also prevents wetting while breathe and cough. To make our work more effective than commercial mask we included the Thymol antibacterial reagent with PVB polymer which prevent the bacteria growth on the mask. Besides, an antibacterial growth study, bacteria filtration efficiency and particle filtration efficiency without affecting the air passage was conducted and a comparative table provided with gram-positive and gram-negative bacteria which is not reported by the researchers yet.
Why PVB was chosen?
Polyvinyl butyral (PVB) has been widely used in many application especially in biomedical application, due to its biodegradability, good adhesion, and solubility in non-toxic solvents making it beneficial in terms of cost effectiveness and eco-friendly characters. Moreover, PVB has excellent dielectric property and non-soluble behavior in moisture environments which helps to form thin nanofiber membranes for moisture prevention and wounds dressing application. These excellent properties fulfills our proposal to form the biodegradable hydrophobic and breathable nanofiber.
The modified part highlighted in main manuscript in page no: 3
Why was Thymol used and not silver particles?
Silver particles have received considerable attention as antimicrobial agents owing to their broad-spectrum of microorganisms. Although, it is high-cost preparation process restrict their usage in commercial low-cost application. Recently, many studies have shown that Thymol has antibacterial and antifungal properties (Refer, 31-34). Thymol incorporation may provide an alternative method in the antibacterial process owing to the phenolic hydroxyl functional group of Thymol.
- The last column in Table 1 is confusing. What does 5wt % PVB (g) mean? Do the authors mean to say that this is the overall weight of the solution?
Answer: Thanks for the reviewer's careful evaluation. We apologize for the blender ratio unit error in the manuscript. We made corrections in our revised manuscript as follows “5 wt% PVB alcohol solution”
- Figure 1 can be improved. I would recommend to only include the construct necessary for the mask and exclude the other things in the image. For example, the image implies that the mask can even stop covid virus. However, no evidence is provided. The size of the virus is much smaller than the bacteria.
Answer: Thanks for the reviewer's deep consideration of our manuscript. We apologize for the inconvenience, the novelty of this work is to prepare the antibacterial hydrophobic fibrous membrane mask. Thymol incorporated PVB nanofiber membrane may not provide the remedial measure to coronavirus but it is able to prevent the disease transmission through water or salivary medium. However, this magnified image can attract more potential researchers and can serve as good reference. Moreover, there is a wider possibility in expanding the research field eventually bringing credits to the publishing journal.
Recent article depicts the same idea of naming their article to attract the target audience.
- Chowdhury, M.A.; Shuvho, M.B.A.; Shahid, M.A.; Haque, A.; Kashem, M.A.; Lam, S.S.; Ong, H.C.; Uddin, M.A.; Mofijur, M. Prospect of biobased antiviral face mask to limit the coronavirus outbreak. Environ Res 2021, 192, 110294, doi:10.1016/j.envres.2020.110294.
- Section 2.3 can be revised. The advantages presented for vertical electrospinning is not unique to vertical setup. It can also be possible for horizontal setup.
Answer: Thanks for the reviewer's comments. Vertical electrospinning can freely adjust the height of the console, relatively easy to meet the diverse needs of people, bring more good application prospect over existing techniques.
- Where was the surfactant added? How was it added and how much of it was added?
Answer: Thanks for the reviewer's comments: Because the electrospun membrane is hydrophobic, in the JIS L1902 antibacterial quantitative test, a surfactant was added to the surface of test bacterial solution. The addition method is to add 5% Tween 80 to the nutrient broth diluted 20 times to adjust the concentration of the bacterial solution, and the final bacterial solution concentration is 2 x 10^5 CFU/mL.

Reviewer 2 Report
Human safety, health management, and disease transmission prevention have become crucial tasks in the present COVID-19 pandemic situation. Particulate matter (PM) pollution, due to industrial development and living environment change, worsens air quality and causes severe health problems, such as respiratory diseases, cardiovascular diseases, and allergies. In the recent years, various influenza and coronavirus diseases have become prevalent, causing harm to human health and large economic losses. Masks are widely available and create a safer and disease transmission–free environment. Besides, masks can prevent droplets and PM from invading the human body and prevent respiratory infections. The types of masks include cotton masks, general medical masks, surgical masks, activated carbon masks, and N95 masks. In this manuscript, this investigation presented a facile method of fabricating masks through electrospinning nontoxic polyvinyl butyral (PVB) polymeric matrix with the antibacterial component Thymol, a natural phenol monoterpene. Based on the results of Japanese Industrial Standards and American Association of Textile Chemists and Colorists methods, the maximum antibacterial value of the mask against gram-positive and gram-negative bacteria was 5.6 and 6.4, respectively. The topic is important, the results are interesting and the methodology followed is appropriate, while the content falls well within the scope of this Journal. The layout is clear and easy to understand. In general the paper makes fair impression and my recommendation is that it merits publication in this Journal, after the following major revision:
- The introduction should be reconstructed to present a coherent literature review. It may help the authors by answering the following questions: Why are these works relevant? Which specific problems were addressed? How are the previous results related with the latest work? What are the outstanding, unresolved, research issues? Answering the questions leads to the novelty of the proposed work naturally.
- Experiment part. Although the results look “making sense”, the current form reads like a simple lab report. The authors should dig deeper in the results by presenting some in-depth discussion.
- In Fig. 2 (d), the authors should give the explanations for the difference of data collected from different sources.
- Thorough optimization of the small-diameter nanofiber–based antibacterial mask led to denser accumulation and improved PFE and pressure difference; the mask was thus noted to meet the present pandemic requirements. The as-developed nanofibrous masks have the antibacterial activity suggested by the National Standard of the Republic of China (CNS 14774) for general medical masks. The authors should give some explanation on above results and data.
- Masks have been widely used in many fields of life. In this research, Thymol/PVB composite nanofibrous membrane was produced with various optimization conditions for the antibacterial mask. Generally, masks are made of fibrous porous media, (see [A fractal model for capillary flow through a single tortuous capillary with roughened surfaces in fibrous porous media, Fractals, 2021, 29(1):2150017; Powder Technology, 2019, 349:92-98]). Authors should introduce some related knowledge to readers.
- Please, expand the conclusions in relation to the specific goals and the future work.
Author Response
We appreciate the comments from the Academic editor and reviewers. The following is our point-to-point response to the comments.
Review-2
Human safety, health management, and disease transmission prevention have become crucial tasks in the present COVID-19 pandemic situation. Particulate matter (PM) pollution, due to industrial development and living environment change, worsens air quality and causes severe health problems, such as respiratory diseases, cardiovascular diseases, and allergies. In the recent years, various influenza and coronavirus diseases have become prevalent, causing harm to human health and large economic losses. Masks are widely available and create a safer and disease transmission–free environment. Besides, masks can prevent droplets and PM from invading the human body and prevent respiratory infections. The types of masks include cotton masks, general medical masks, surgical masks, activated carbon masks, and N95 masks. In this manuscript, this investigation presented a facile method of fabricating masks through electrospinning nontoxic polyvinyl butyral (PVB) polymeric matrix with the antibacterial component Thymol, a natural phenol monoterpene. Based on the results of Japanese Industrial Standards and American Association of Textile Chemists and Colorists methods, the maximum antibacterial value of the mask against gram-positive and gram-negative bacteria was 5.6 and 6.4, respectively. The topic is important, the results are interesting and the methodology followed is appropriate, while the content falls well within the scope of this Journal. The layout is clear and easy to understand. In general the paper makes fair impression and my recommendation is that it merits publication in this Journal, after the following major revision:
- The introduction should be reconstructed to present a coherent literature review. It may help the authors by answering the following questions: Why are these works relevant? Which specific problems were addressed? How are the previous results related with the latest work? What are the outstanding, unresolved, research issues? Answering the questions leads to the novelty of the proposed work naturally. Why are these works relevant? Which specific problems were addressed? How are the previous results related with the latest work? What are the outstanding, unresolved, research issues?
Answer: Thanks for the reviewer's deep consideration of our manuscript. The N95 commercial medical mask offers good bacteria filtration efficiency than the others although they cannot be wear for a long time because of their favourable adhesion property. Moreover, melt-blown fabrics diameter range is approximately <2 m (Table S8, Figure S2) which is not suitable for very tiny particles because, if fiber diameter is high the accumulation of fiber at a specific area would be less. Therefore, we employ facile electrospinning process to optimize the fiber diameter to meet CNS 14774 specification for filtration. We also estimated air penetration test in order to avoid breathing difficulties. The hydrophobic property of PVB not only protect us from water droplet environment it also prevents wetting while breathe and cough. To make our work more effective than commercial mask we included the Thymol antibacterial reagent with PVB polymer which prevent the bacteria growth on the mask. Besides, an antibacterial growth study, bacteria filtration efficiency and particle filtration efficiency without affecting the air passage was conducted and a comparative table provided with gram-positive and gram-negative bacteria which is not reported by the researchers yet
- Experiment part. Although the results look “making sense”, the current form reads like a simple lab report. The authors should dig deeper in the results by presenting some in-depth discussion.
Answer: Thanks for the reviewer's suggestions. We are made some correction as per your suggestions. The Corrected part highlighted in main manuscript.
Experimental part:
“A known weight of test samples were added to observe whether the Petri dish had a zone of inhibition, we measured the width of the sample (D) and the sum of the width of the sample and the zone of inhibition (T); then calculated the width of the zone of inhibition (W) with the following formula (1), and the result is expressed as an average 0.1 mm”
The modified part highlighted in main manuscript in page no: 4
“Following the procedure of CNS 14775, the Bacterial Filtration Efficiency (TTRI, TW) of the synthesized mask was calculated. A formed bacteria (S. aureus biological) aerogel size is approximately (3.0 ± 0.3) μm), which is controlled by the nebulizer system. The percentage of bacteria absorption before and after filtering through the mask and rate of applied pressure was estimated. The results were subjected to the following equation to calculate the percentage of BFE. Besides, the same procedure was followed for the 30 commercial masks to compare the obtained results”
The modified part highlighted in main manuscript in page no: 5
Result and Discussion:
“Although, as shown in Table 2 and Figure S1, spraying the tested bacteria liquid on the nanofibrous membrane or net-like architecture did not form any contact with the substrate. Even though applying gentle pressure using a glass plate, this lack of interface formation results to fail the formation of the inhibition zone. In contrast, Group A and Group C showed the formation of zone inhibition on alumina foil with a high concentration ratio of Thymol in PVB (1:1). However, the width of the inhibition zone increased slightly, the difference was not large”
The modified part highlighted in main manuscript in page no: 6.
“Therefore, in this study, 30 commercial masks were tested and analyzed to determine the correlation between their BFE and PFE. In addition, a Thymol/PVB nanofibrous membrane was prepared through electrospinning with the Group F parameters, with fiber collection lasting 1–6 h, it was observed that by increasing the nanofiber collecting time leads to the higher areal density nanofiber membrane which helps to trap the bacteria aerosol furthermore these BFE results were compared with the PFE data’s. As shown in Table S9, the correlation between BFE and PFE for the different masks were nonlinear, although when the PFE was exceed 73%, the BFE reach its maximum. Therefore, in this study we prepared the mask with aim of achieve the PFE of ≥80% and then conducted the BFE test. In theory, if the BFE value exceed 95% which could meet the CNS 14774 specifications.”
The modified part highlighted in main manuscript in page no: 12
- In Fig. 2 (d), the authors should give the explanations for the difference of data collected from different sources.
Answer: Thanks for the reviewer's comments. Here, we performed an antibacterial test with all three bacteria's with various percentage of Thymol contained PVB nanofiber. Ratio, 0.1. 0.2, 0.4 show an antibacterial effect on gram-negative even though which is not meet the JIS L1902 requirements for gram-positive bacteria. Among them, the 0.6:1 ratio exhibit strong action against overall.
- “Thorough optimization of the small-diameter nanofiber–based antibacterial mask led to denser accumulation and improved PFE and pressure difference; the mask was thus noted to meet the present pandemic requirements. The as-developed nanofibrous masks have the antibacterial activity suggested by the National Standard of the Republic of China (CNS 14774) for general medical masks” The authors should give some explanation on above results and data.
Answer: Thanks for the reviewer comments. Novelty of this work to prepare the antibacterial hydrophobic nanofiber membrane for the filtering mask with the help of vertical electrospinning. Generally in medical mask preparation reduction of fiber diameter would led to smaller pore nanofiber membrane, which effectively trap the molecular size particle as well as aerosol. However, this is not suitable in all the case mainly it would cause the breathing issue due to the adhesion. Therefore, optimization step is most important. Considering the above bottlenecks, we herein optimize the electrospinning parameter by pressure passage and particle filtration efficiency characterizations. (Table 4. Figure 5).
Therefore, we added the new reference to the existing flow in the page No: 12.
- Yim, W.; Cheng, D.; Patel, S.H.; Kou, R.; Meng, Y.S.; Jokerst, J.V. KN95 and N95 Respirators Retain Filtration Efficiency despite a Loss of Dipole Charge during Decontamination. ACS Appl Mater Interfaces 2020, 12, 54473-54480, doi:10.1021/acsami.0c17333.
- “Masks have been widely used in many fields of life. In this research, Thymol/PVB composite nanofibrous membrane was produced with various optimization conditions for the antibacterial mask” Generally, masks are made of fibrous porous media, (see [A fractal model for capillary flow through a single tortuous capillary with roughened surfaces in fibrous porous media, Fractals, 2021, 29(1):2150017; Powder Technology, 2019, 349:92-98]). Authors should introduce some related knowledge to readers
Answer: Thanks for the reviewer's suggestion. We added the new reference to the existing flow to support our discussion in page 12.
“Moreover, to support this many other researchers also have been investigated the filtration efficiency using the analytical methods on the porous fiber membranes. [52, 53]”
- Xiao, B.; Huang, Q.; Chen, H.; Chen, X.; Long, G. A Fractal Model for Capillary Flow through a Single Tortuous Capillary with Roughened Surfaces in Fibrous Porous Media. Fractals 2021, 29, 2150017, doi:10.1142/s0218348x21500171.
- Xiao, B.; Wang, W.; Zhang, X.; Long, G.; Fan, J.; Chen, H.; Deng, L. A novel fractal solution for permeability and Kozeny-Carman constant of fibrous porous media made up of solid particles and porous fibers. Powder Technology 2019, 349, 92-98, doi:10.1016/j.powtec.2019.03.028.
- Please, expand the conclusions in relation to the specific goals and the future work.
Answer: Thanks for the reviewer's suggestion. We polished the conclusion to our specific goal in the page (13-14).
“Extension of industries may address their countries growth. Although, it brings an inevitable side effect to their people. Especially, food and airborne disease cause some serious issue. The face masks are an affordable and effective safety measure. Cost-effective bio-degradable eco-friendly masks are inviting field of research due to increasing demands on healthcare and raising concerns of personal hygiene and safety. Through this research, we opened up a low-cost biodegradable face mask fabrication process which could protect and sustain the human living”
